# Heart Failure in Menopause: Treatment and New Approaches

**DOI:** 10.3390/ijms232315140

**Published:** 2022-12-01

**Authors:** Jaqueline S. da Silva, Tadeu Lima Montagnoli, Mauro Paes Leme de Sá, Gisele Zapata-Sudo

**Affiliations:** 1Programa de Pós-Graduação em Cardiologia, Instituto do Coração Edson Saad, Hospital Universitário Clementino Fraga Filho, Universidade Federal do Rio de Janeiro, Rio de Janeiro 21941-902, Brazil; 2Programa de Pós-Graduação em Farmacologia e Química Medicinal, Instituto de Ciências Biomédicas, Universidade Federal do Rio de Janeiro, Rio de Janeiro 21941-902, Brazil; 3Instituto do Coração Edson Saad, Faculdade de Medicina, Universidade Federal do Rio de Janeiro, Rio de Janeiro 21944-590, Brazil

**Keywords:** heartfailure, diastolic dysfunction, menopausal, estrogen, therapeutic targets

## Abstract

Aging is an important risk factor for the development of heart failure (HF) and half of patients with HF have preserved ejection fraction (HFpEF) which is more common in elderly women. In general, sex differences that lead to discrepancies in risk factors and to the development of cardiovascular disease (CVD) have been attributed to the reduced level of circulating estrogen during menopause. Estrogen receptors adaptively modulate fibrotic, apoptotic, inflammatory processes and calcium homeostasis, factors that are directly involved in the HFpEF. Therefore, during menopause, estrogen depletion reduces the cardioprotection. Preclinical menopause models demonstrated that several signaling pathways and organ systems are closely involved in the development of HFpEF, including dysregulation of the renin-angiotensin system (RAS), chronic inflammatory process and alteration in the sympathetic nervous system. Thus, this review explores thealterations observed in the condition of HFpEF induced by menopause and the therapeutic targets with potential to interfere with the disease progress.

## 1. Introduction

Cardiovascular diseases (CVD) are still the leading causes of death worldwide [1,2] which includes heart failure (HF) with high morbidity [3] and a survival rate of only 50% after five years of diagnosis [4]. The 2019 Heart Disease and Stroke Statistical Update indicates that the reported incidence of CVD was 77.2 and 78.2% for men and women, respectively aged 60–79 years and increased to 89.3 and 91, 8% when over 80 years [5]. Since aging is an important risk factor for CVD, elderly women have an increased incidence of HF with preserved ejection fraction (HFpEF), with a risk rate for hospitalization of 2.46 and 5.22 in the 60–69 and 70–79 age group, respectively [6].

Ejection fraction (EF) is an important parameter to determine the prognosis and the response of patients to treatment of HF. HF with reduced EF (HFrEF) is identified in patients with EF ≤40%.However, when with symptoms but with normal systolic function the patients are classified with HFpEF, (EF ≥50%) [3]. Approximately half of diagnosed patients have HFpEF [7] and its prevalence is increasing [3]. HFpEF is more common in women and most of them have associated risk factors, including hypertension, obesity, hyperlipidemia and diabetes mellitus [8]. Thus, advanced age associated with comorbidities contributes to the accelerated rate of morbidity and mortality in women [9]. HFpEF in women has been related to the loss of cardioprotection due to the reduction in estrogen levels induced by menopause. Thus, this review seeks to indicate some of the mechanisms that are involved in HFpEF in women indicating possible targets for therapy, as well as addressing the most recent clinical approaches for this population.

### Heart Failure in Menopause and Therapeutic Targets

According to the guidelines of 2021 European Society of Cardiology, HF is a clinical syndrome consisting of symptoms including shortness of breath, ankle swelling, and fatigue that can be accompanied by signs such as elevated jugular venous pressure, edema. HF is due to a structural and/or functional abnormality of the heart that results in elevated cardiac pressures and/or inadequate cardiac output at rest and/or during exercise. Most commonly, HF is due to myocardial dysfunction: systolic, diastolic, or both and has been divided into distinct phenotypes based on the measurement of left ventricular EF: reduced, HFrEF, mildly reduced (HFmrEF—EF between 41–49%) and preserved, HFpEF [10]. Patients with HFmrEF have, on average, characteristics more similar to HFrEF than HFpEF, being more common in young men and more likely to have coronary artery disease (CAD) [11,12,13] and less likely to have atrial fibrillation and comorbidities. However, outpatients with HFmrEF have lower mortality than those with HFrEF. HFpEF differs from HFmrEF and HFrEF because it is more often in elderly women and is associated with atrial fibrillation, chronic kidney disease, and non-cardiovascular comorbidities [14].

Sex differences that lead to discrepancies in risk factors and development of CVD reduce with aging, and are largely attributed to the low level of circulating estrogen [15]. The prevalence, pathophysiology, and mortality rates associated with CVD differ between men and women according to age [16]. Premenopausal women have lower CVD prevalence and mortality compared to men, while postmenopausal CVD prevalence becomes similar in both sexes [16], directly related to the reduction of the cardioprotective effect of estrogen. Because of higher life expectancy of women, the prevalence of CVD is equal in male and female elderly population but with higher mortality in women due to a low level of estrogen. It is usually the initial appearance of coronary heart disease in men, while women have stroke or HF as the first event [17].These differences in CVD incidence between men and women decrease with increasing age [18], coinciding with estrogen depletion observed in post-menopausal women.

In the cardiovascular system, the activation of estrogen receptors (ER) causes activation of several signaling pathways that promote cardioprotection, including reduction of cellular hypertrophy, apoptosis and inflammatory responses, as well as promoting antioxidant actions. Estrogen exerts its actions by activating ERα and ERβ, classically nuclear receptors, and G protein-coupled receptor (GPER) which has membrane distribution with rapid signaling [19].

Estrogen binding to ERα or ERβ in its monomeric form promotes homo or heterodimerization, nuclear translocation and association with co-regulators of gene transcription (canonical signaling) [20,21] and indirectly regulates gene expression through interaction with c-Jun and c-Fos [21].

Activation of ERα promotes phosphorylation of Akt and endothelial nitric oxide synthase (eNOS), with consequent activation of cyclic guanosine monophosphate (cGMP) dependent protein kinase (cGMP-PKG) signaling pathways, promoting vasodilation and attenuating hypertrophy and pathological cardiac apoptosis. Reduction of cardiac hypertrophy occurs by increasing degradation of calcineurin and inhibition of its activity. Apoptosis is inhibited by estrogen through the activation of Akt, the induction of transcription of superoxide dismutase (SOD), the promotion of antioxidant action and the reduction of tumor necrosis factor α (TNF-α) expression [19].

ERβ promotes activation of phosphatidylinositol 3-kinase (PI3K) and protein kinase A (PKA) and is involved in the negative modulation of inflammatory and fibrotic responses [19,21,22].

Depending on the cell type, the GPER activates several signaling pathways: (1) Gα protein-dependent cell signaling, promoting activation of PKA and phospholipase C (PLC), which will promote deactivation of raf-1 proto-oncogene, serine/threonine kinase (raf-1) and mobilization of intracellular Ca^2+^, respectively [9]; (2) c-Src and induces an increase in matrix metalloproteinase (MMP) expression; and (3) the mitogen-activated protein kinase (MAPK), PI3K, extracellular signal-regulated kinase (ERK) and Akt [19,23]. These actions will contribute to the vasorelaxation, inhibition of proliferation in vascular smooth muscle cells and cardiac fibroblasts [19,22]. GPER activation also promotes anti-inflammatory effects by downregulation of interleukin (IL)-6 expression in macrophages and repressing the activity of nuclear factor kappa B (NF-κB), through modulation of kinase inhibitor κB phosphorylation (IKK) and c-jun N-terminal (JNK) [24]. GPER is also involved in Ca^2+^ homeostasis via interaction with beta 2 adrenergic receptors (β2AR) [25].

Thus, ER adaptively modulates fibrotic, apoptotic, inflammatory processes and calcium homeostasis, factors that are directly involved in the HFpEF and therefore, during menopause, the estrogen depletion reduces this cardioprotection.

HFpEF consists of normal contractile function with [8] diastolic dysfunction which is considered a functional impairment in relaxation or filling of the left ventricle (LV) [6]. This condition could be considered a normal aging process, but sustained chronic diastolic dysfunction is a risk factor for HF [8,26]. HFpEF is predominantly manifested with reduced diastolic relaxation and LV compliance, with minimal loss of cardiomyocytes [27]. Furthermore, it has been observed that the presence of the inflammatory process, coronary endothelial dysfunction, interstitial fibrosis and impaired cardiomyocyte relaxationfavor LV remodeling and diastolic dysfunction [27]. Women are more likely to develop diastolic dysfunction and consequently HFpEF, with hypertrophy and a smaller LV cavity [28]. These anatomical changes predispose elderly women to diastolic dysfunction with consequent worsening of relaxation, increased filling pressure and ventricular stiffness [28]. Thus, elderly women have increased rate of vascular stiffness, which leads to endothelial dysfunction, arterial hypertension, and microvascular disease [29], contributing to the HFpEF [6].

## 2. Mechanism of Menopause-Induced HFpEF and Targets for Treatment

In an attempt to investigate HFpEF, several animal models have been used including the bilateral oophorectomy (OVX) in rats, which provides a valuable model of hypertrophy and HF and allows the investigation of the role of estrogen-induced cardioprotection [30]. Comorbidities associated with female aging were evaluated in the OVX model which was also been linked to high blood pressure [31,32,33,34,35,36,37,38,39], volume overload [40,41], metabolic syndrome [42,43,44], type 2 diabetes (T2DM) [45,46,47,48,49] and stress [25,50,51].

Several signaling pathways are closely involved in the development of HFpEF during menopause, including dysregulation of the renin-angiotensin system (RAS), chronic inflammatory process and alteration in the sympathetic nervous system (Figure 1).

### 2.1. Renin-Angiotensin System

RAS is a central regulator of cardiovascular function and plays a key role in the pathophysiology of HF. The generation of angiotensin II (Ang II) mediated by angiotensinogen-converting enzyme (ACE) and chymase is closely associated with cardiac remodeling and dysfunction. A decrease in type 2 ACE (ACE2) activity can alter the balance of local cardiac RAS interfering with the development of diastolic dysfunction [52,53]. In a model of congenital hypertension, OVX-induced estrogen depletion modulates remodeling and promotes cardiac dysfunction which are combined with increased expression of Ang II, chymase and number of cardiac mast cells, a condition that is attenuated with replacement of 17β-estradiol (E2) [31,35].

In cardiac cells, the components of RAS are fibroblasts, endothelial cells, myocytes and macrophages. These local systems promote the production of Ang II that interacts with intracellular Ang II receptor type 1 (AT1) to exert its effects [54]. Activation of an intracellular Ang-(1–12)/Ang II/chymase system is involved in pathophysiological alterations in CVD.

Cardiac myocyte chymase activity is associated with diastolic function and cardiac structure, while there are no relevant functional and structural relationships between ACE or Ang(1–7) formation pathways [32]. Chymase, a serine protease found in granules of mast cells and non-cardiac myocytes [55], is released into the cardiac interstitium after injury or inflammation. Chymase acts within the ACE-independent Ang II formation pathway and activates the transforming growth factor beta (TGF-β) and other promoters of extracellular matrix degradation, thus playing a role in tissue remodeling [52,53]. E2 prevents cardiac remodeling in OVX submitted to volume overload, by inhibiting the release of chymase by mast cells [40]. Normotensive females undergoing OVX show increased gene expression and enzyme activity of cardiac chymase and a modest but significant reduction in cardiac ACE2 activity which could explain the worsening of diastolic function because of increased filling pressure and reduced cardiac relaxation (Figure 2). In contrast, diastolic dysfunction in hypertensive female rats with OVX (SHR-OVX) appears to be associated with reduced cardiac ACE2 activity [33]. Modulation of the RAS was also demonstrated in OVX mice with reduced cardiac expression of ACE2, Mas receptor and angiotensin receptor type 2 (AT2), as well as an increase in AT1 expression, but no change in ACE expression [56].

Cardiac fibroblasts have a large amount of chymase mainly during stress and volume overload [55] and local Ang II enhances cardiomyocyte responses [57]. Human tissue expresses only the alpha isoform of chymase, which is also one of the isoforms found in rodents. Five of the ten mouse mast cell proteases (rMCP-1, rMCP-2, rMCP-3, rMCP-4 and rMCP-5) exhibit chymase activity [58]. Rat rMCP-5 would be equivalent to human α chymase [59]. The expression of mRNA for rMCP-1 and rMCP-5 in normotensive or hypertensive rats increases after OVX. The activity of chymase involved in the formation of Ang II from Ang(1–12), correlates with mRNA expression of rMCP-1 and rMCP-5 and also with diastolic dysfunction induced by estrogen depletion [34]. Localization-specific chymase-targeted inhibition could be more effective in the treatment of postmenopausal HF than non-specific ACE inhibition. Ang II modulates cardiac fibroblasts, inducing fibrosis through activation of PKA and 5′ adenosine monophosphate-activated protein kinase (AMPK) and Rho kinase (ROCK). ER activation by E2 or a specific ERβ agonist (βLGND2) inhibits the Ang II/PKA/AMPK/ROCK pathway in isolated cardiac fibroblasts by reducing TGF-β expression, MMP 2 and 9 expression and activity, and production and connective tissue growth factor (CTGF) function, as well as the conversion of fibroblasts to myofibroblasts with reduced collagen production (Figure 2). Similarly, E2 or βLGND2 treatment also reduces the expression of TGF-β, AMPK and CTGF in OVX. Thus, E2 acting on ERβ would inhibit the cardiac hypertrophic and fibrotic effects of Ang II [60]. However, in the condition of estrogen-depleted hypertensive SHR-OVX [35], the treatment with an agonist of the ERβ receptor, DPN, does not improve diastolic dysfunction. The differences between the experimental models and the selectivity/potency of the agonists used could explain the divergences of findings. The selectivity of βLGND2 is approximately 25 times for the activation of ERβ over ERα [61], while DPN is 70-fold for the activation of ERβ over ERα [62], suggesting that βLGND2 may not be exclusively activating ERβ.

Recently, it was demonstrated a possible involvement of GPER in the inhibitory effect of Ang II in fibroblasts. In cardiac fibroblasts culture, G1 inhibits Ang II-induced proliferation and apoptosis. In addition to this, in a model of Ang II-induced atrial fibrillation in OVX mice, the treatment with G1 reduces fibrosis and atrial fibrillation with alteration in the expression of proteins and genes related to apoptosis and involved in inflammation, resulting in an upregulation for mothers against decapentaplegic homolog(Smad)7 and inhibition of the TGF-β/Smad pathway [63].

GPER has a unique role in the maintenance of diastolic function and in the modulation of the blood pressure. Treatment with an agonist of GPER or ERα provides improvement in diastolic function after estrogen depletion when compared with an untreated animal because of the reduction in ACE activity, AT1 and cardiac chymase expression. Although PPT improves diastolic function in SHR-OVX, it does not induce changes in local RAS [35].

Combination therapy consisting of G1 and ACE inhibitor lisinopril (G1-Lys) may act synergistically to increase the balance of circulating angiotensin peptide towards Ang-(1–7), contributing to advantageous diastolic function. G1 or lisinopril alone improve diastolic dysfunction in SHR-OVX, with increased cardiac expression of Mas (Mas-R) and ACE2 receptors and shift the circulating RAS towards the formation of Ang-(1 7), effects that are amplified in G1-Lys combination therapy. Furthermore, the G1-Lys normalizes the blood pressure of SHR-OVX without causing hypotension, indicating that the blood pressure control of SHR-OVX promoted by lisinopril favors the cardioprotective effect of G1. Thus, G1-Lys therapy would be considered for the treatment of postmenopausal cardiac dysfunction in the presence of pre-existing hypertension [36].

The ACE inhibitor, enalapril, improves cardiac function and structure in both young SHR and elderly normotensive rats (82 weeks) undergoing OVX [37,38]. In addition, enalapril reduces coronary bed reactivity and dobutamine-induced contractile response in elderly OVX rats. Although an improvement in cardiac function was also observed using aerobic physical training, the combination of enalapril did not promote important additional effects [38]. These results are different from those observed by Dutra et al., (2017) because enalapril exacerbated cardiac fibrosis and sympathetic participation in determining baseline heart rate, without significantly affecting cardiac functionin young OVX females [64]. Thus, it seems that ACE inhibition in order to favor improvement in cardiac dysfunction induced by estrogen depletion may be dependent on the age of the animals and total time of treatment.

Increased chymase and reduced local ACE2 activity are evidently involved in menopausal diastolic dysfunction by modulating cardiac remodeling. The increase in Ang II promotes the modulation of fibroblasts for fibrotic, proliferative and inflammatory condition. Activation of ERs, mainly GPER and ERβ, could act to minimize the deleterious effects of RAS in menopause. Selective blockade of the local RAS and early decision to start therapy seem to be a promising approach for the treatment of HFpEF during menopause.

### 2.2. Inflammation, Metabolic Syndrome and Diabetes

During menopause, in many cases, metabolic syndrome has been described to be related to the development of CVD [65,66,67] and inflammation would be the link between metabolic syndrome and HFpEF. The main source of pro-inflammatory cytokines in metabolic syndrome are adipocytes and macrophages, where the latter may have a change in their phenotype that makes them more pro-inflammatory and together with an overproduction of cytokines promote local inflammation and subsequently propagate to the systemic level. The production of cytokines IL-1β, IL-6 and TNF-αplays a positive role promoting a long-term inflammatory state, mainly through the activation of transcription factors such as NF-κB, JNK and signal transducer and activator of transcription (STAT) [68].

The association of OVX with a hypercaloric diet (OVX-High Fat), in addition to promoting weight gain that is already observed during menopause, intensifies the increase in levels of total cholesterol, triglycerides and plasma glucose and the expression of TNF-α, NF-κB [42]. Fruits rich in polyphenolic compounds (anthocyanins and flavonoid glycosides) have an antioxidant effect which could be beneficial to improve metabolic parameters and cardiovascular risk in post-menopause [69]. Thus, those compounds decrease the metabolic syndrome by increasing the expression of peroxisomal proliferation-activated receptor γ (PPAR-γ) that could be acting to reduce inflammatory processes [42]. OVX-High Fat causes a significant increase in cardiac collagen content and reduces cardiac levels of nitrotyrosine (3-NT) and glutathione (GSH) [43]. Physical exercise could be an additional alternative for the prevention and treatment of HF in these cases, because it can attenuate cardiac fibrosis induced by OVX-High Fat via regulation of GSH/3-NT and MMP-2. The mechanism related to this exercise-induced cardioprotection is due to the favorable profile of MMP/tissue inhibitor of metalloproteinases (TIMP) resulting in activation of MMP [43] which can be activated by proteolytic and non-proteolytic pathways. Proteolytic activation can occur by post-translational modification caused by peroxynitrite (ONOO^-^) in the presence of GSH [70]. Thus, exercise-induced MMP-2 activation improves the balance between MMP and TIMP, contributing to cardioprotection, acting positively on OVX-induced cardiac remodeling [43].

Estrogen depletion in hypertensive female rats exacerbates inflammation-related protein levels of TNF-α, NF-κB, cyclooxygenase 2 (COX-2), inducible nitric oxide synthase (iNOS) and IL-6,fibrotic-related protein levels of TGF-β, Smad2/3, CTGF, tissue-type plasminogen activator (tPA), MMP-9, and collagen I. Treadmill physical training may prevent inflammation and cardiac fibrosis of this condition possibly by decreasing AT1 [39].

Recently, the member of the sirtuin protein family (SIRT) SIRT1 has been highlighted for its antioxidant and anti-inflammatory properties [71,72,73] and the increase in its expression has been demonstrated in OVX treated with melatonin alone or in combination with hormone replacement therapy (TRH) or physical training. Increased IL-10 expression and reduced IL-6 and TNF-α level can be responsible for the increase inSIRT1, which can improve oxidative damage and inflammation in the heart and aorta of OVX rats. SIRT1could be considered as a therapeutic target for cardiac dysfunction in postmenopausal women [74].

T2DM-induced cardiac dysfunction is associated with increased cardiac morbidity and mortality which is greater in women than in men [75]. The main molecular mechanisms for the development of diabetic cardiomyopathy include the NF-κB pathway and the RAS, in addition to the accumulation of advanced glycation end products (AGE) and overexpression of inflammatory cytokines. Chronic inflammation, structural and metabolic alterations including alterations in calcium homeostasis, cardiomyocyte apoptosis, hypertrophy and myocardial fibrosis contribute to HF. Increase in pro-inflammatory cytokines, such as TNF-α and IL-6, IL-8, IL-1β and C-reactive protein (CRP) are closely related to fibrosis and HF which is a final stage ofT2DM [76].

OVX-induced menopause associated with T2DM (OVX-DM2) in rats induces an exacerbation of the metabolic disorder possibly due to increased insulin resistance, with intense atherogenic status and cardiac hypertrophy due to an increase in Ang II and an imbalance in favor of the inflammatory cytokines TNF-α and IL-6 over IL-10. However, treatment with E2 only partially restores the changes observed in OVX-DM2. Moreover, the combined treatment of E2 with progesterone in OVX-DM2 is ineffective, abolishing the cardioprotective effect of E2 [46]. Antagonists of ERα (MPP) and ERβ (PHTPP) reinforce metabolic disorders and cardiac inflammation, with an increase in TNF-α and IL-6 and a decrease in IL-10 levels. These indicate that classic ERs also have cardioprotective effects in diabetes [47]. An increase in cardiac fibrosis and apoptosis related to increased expression of iNOS observed in OVX-DM2 can be reduced after activation of GPER using G1 or after specific iNOS inhibition with W1400 which reduces the expression of iNOS, fibrosis and cardiac apoptosis. Thus, GPER activation could inhibit myofibroblasts in OVX-DM2 rats, suppressing cardiac iNOS activity and, consequently, nitric oxide (NO) levels [48].

Considering most OVX studies which show a detrimental effect of hormone depletion on glycemic and lipid control [46,47,48,77,78], the estrogen depletion induced by OVX in an obese-diabetic rat model (Zucker Diabetic Fatty—ZDF) attenuates both diabetes and cardiac dysfunction, explained by the lack of cardioprotective profile of E2 [78]. E2 exacerbates myocardial dysfunction in ZDF rats by disrupting adiponectin signaling at cardiac level. In part, through the reduction of connexin-43 and phosphorylated survival molecules, ERK1/2 and phosphorylated AKT regulation in the heart [45], activation of pro-inflammatory pathways in the paraventricular nucleus occurs through the upregulation of ERα/ERβ receptors and adenosine A1/A2a receptors [49]. E2 paradoxically transforms into a pro-inflammatory hormone in the presence of oxidative stress [79], such as that induced by diabetes and obesity [80]. The cardiac dysfunction would be closely related to estrogen levels, since a short term of OVX-DM2 (4 weeks) does not induce cardiac dysfunction. However, when OVX-DM2 are treated with E2, cardiac complications and mortality are similar to those observed in young females with T2DM. Reduction in AKT and ERK1/2 phosphorylation due to connexin 43 reduction are E2-dependent in OVX-DM2 since these proteins are not altered in estrogen depletion condition [45]. The proteins, AKT and ERK1/2, have been associated with protection of cardiomyocytes against apoptosis induced by oxidative stress [81], and their reduction would therefore contribute to the development of cardiac dysfunction. There is an increase in cardiac ERα expression in the presence of T2DM, but it is higher when treated with E2 [45], which correlated with reactive oxygen species, TNF-α and phospho- death-associated protein kinase -3 levels [49]. The pro-inflammatory changes related to cardiac dysfunction may be mediated by adenosine A1 and A2 receptors that are elevated in E2-treated OVX-DM2 [82,83]. A blockade of ERα and, to a lesser extent, ERβ, inhibits the exacerbation of oxidative stress in rats treated with E2 [82], and blockade of adenosine A1 and A2 attenuates the cellular inflammation [84,85].

It is important to consider that a non-selective activation of the ER using HRT in menopausal women with HFpEF together with metabolic syndrome or T2DM is not an appropriate alternative. However, a more selective treatment regarding the activation of ER or the inhibition of inflammation and fibrosis pathways could be highly beneficial for these patients.

#### Sympathetic Nervous System

Normal estrogen levels protect cardiomyocytes from stress mediated by the activation of β2AR [86]. In OVX rats, the stress induced by adrenaline promotes aggravation of arrhythmias and a decrease in cardiac contractility. A blockade of the β2AR-Gs pathway increases cardiomyocyte damage similar to what happens in OVX, suggesting that this pathway is less active in the condition of estrogen depletion. Thus, estrogen depletion reduces β2AR-Gi activation, which in turn exacerbates the damage to myocytes and increases arrhythmia susceptibility (Figure 3).

Thus, estrogen in a normal condition increases the activity of β2AR-Gi, but in a state of stress, it is responsible for increasing the activity of both β2AR-Gi and β2AR-Gs, promoting cardiac protection during stress. However, in the presence of estrogen depletion, there is an abnormal coupling of the β2AR-Gs/Gi pathways during stress, promoting arrhythmias and cardiac injury [50]. A normal level of estrogen decreases plasma levels of noradrenaline and adrenaline, inhibiting sympathetic activity and decreasing the level of circulating catecholamine [87,88].

Activation of GPER by E2 improves the contractile and Ca^2+^ regulation in cardiomyocytes from OVX mice [25]. OVX induces reduction of calcium current density and cAMP concentration, which is consequent of increased hydrolysis by phosphodiesterase (PDE) 3A, 3B, 4A and 4D. These alterations are abolished when OVX cardiomyocytes are treated with E2 or in the presence of PDE inhibitor, IBMX. E2 effects are related to the activation of GPER and not of ERα and ERβ, since only the GPER antagonist, G15, blocks the effects of E2 in OVX cardiomyocytes [25]. During an isoproterenol-induced stress period, contraction and transient reduction of Ca^2+^ and cyclic adenosine monophosphate (cAMP) concentration occur in OVX cardiomyocytes, while pre-treatment with E2 and G1 restores these responses. E2 effects are attributed to activation of β2AR, because in knockout animals for β2AR they showed responses similar to OVX. In addition, the receptor expression is reduced on cardiomyocytes from OVX, and is normalized when pre-treated with E2. However, it is important to emphasize that the reduced responses regarding contractility, Ca^2+^ transient amplitude and cAMP level in cardiomyocytes due to estrogen depletion may be dependent on the time course of OVX [89]. Myocardial Ca^2+^ homeostasis is regulated by cAMP/PKA, even in the absence of β-adrenergic stimulation [90]. OVX promotes sarcoplasmic reticulum Ca^2+^ overload in ventricular myocytes [91] resulting from activation of the cAMP/PKA pathway with high Ca^2+^ transients, excitation-contraction coupling (EC). Intracellular cAMP level is a result of the balance of its production via adenylate cyclase and its degradation via PDE, which may be increased in OVX [89]. Since the dysregulation in Ca^2+^ homeostasis is closely linked to the development of cardiac dysfunction, modulation of the cAMP/PKA pathway could be a target for the treatment of menopausal-induced HF. The increase in Ca^2+^ transient in cardiomyocytes may be influenced by the expression of proteins related to Ca^2+^ storage in the sarcoplasmic reticulum [92]. Regardless of estrogen levels, there is a difference in the expression of proteins related to intracellular Ca^2+^ storage in cardiomyocytes of male and female mice. The ryanodine receptor (RyR2) mRNA expression is higher in females than in males, but males have a higher expression of phospholamban(PLB) mRNA and sarcolipin (SLN) [25]. Estrogen depletion promotes reduction of RyR2 and SNL mRNA, but increases levels of PLB mRNA which correlates with lower Ca^2+^ transient amplitude and contraction possibly due to lower sarcoplasmic reticulum Ca^2+^ release [25].

Estrogen could also modulate the β2AR-Gs/Gi pathway in macrophages and E2 deficiency allows the induction of chronic stress-induced cardiomyopathy characterized by cardiac dysfunction, maladaptive myocardial hypertrophy, unresolved pro-inflammatory responses, and fibrosis. The presence/supplementation of E2 during stress avoids all adverse effects of chronic stress, while preventing excessive β2AR depletion. Furthermore, E2 facilitates the resolution of the myocardial inflammatory response by facilitating a reparative response [51].

There is a reduction in noradrenaline content in the cardiac tissue of OVX rats but, there is an increase in the expression of the noradrenaline transporter (NET), the β1 adrenergic receptor (β1AR) and the Ca^2+^–dependent protein calmodulin kinase II (CAMK II) that is reversed after treatment with E2 [93]. In the absence of estrogen induced by OVX, there is an asymmetry in the expression of noradrenaline synthase, dopamine β-hydroxylase and NET because of an impairment of NET function. The reduced number of NET binding sites and noradrenaline release may produce a consequent decrease in noradrenaline content [94,95]. Thus, E2 not only regulates the expression of noradrenaline-related proteins, but also directly affects the function of NET uptake and this mechanism appears to be involved in the development of CVD in menopause [93]. The increase β1AR and CAMKII expression could be responsible for the hypertrophy, cardiac structural damage and HF present in OVX (Figure 3).

## 3. Treatment of Women with Menopausal-Induced HF

### 3.1. Pharmacological Therapy

In general, the pharmacological clinical trials that may help to establish new guidelines for the treatment of HFpEF are not specific for menopausal-induced HF. However, much has advanced as these trials have currently included a significant number of women which allows to make some considerations regarding this specific group of patients (Table 1).

Modulation of the RAS for the treatment of HFpEF has shown mostly neutral results for drugs such as perindoprol, candesartan, irbesartan, spironolactone, and sacubitril/valsartan (Sac/Val) [10,96]. In regard of the analysis of the Aldosterone Antagonist Therapy for Adults with Heart Failure and Preserved Systolic Function (TOPCAT—NCT00094302), there were no differences between the sexes using placebo or spironolactone for the primary outcome or its components. However, there was a reduction in all-cause mortality associated with spironolactone therapy in women, with significant interaction between sex [97]. A comparative study between the use of valsartan and Sac/Valin patients with HFpEF in which approximately 50% were female showed a potential benefit for this specific group [98]. Many clinical studies related to the use of Sac/Val to treat HFpEF evaluate: (1)the primary outcome as a composite of total hospitalizations (first and recurrent) and death from cardiovascular causes, myocardial infarction, or stroke, all-cause mortality, and renal outcome (decreased glomerular filtration, development of end-stage renal disease, or death from renal failure) and (2)the secondary outcome including change in Clinical Summary Score of the Kansas City Cardiomyopathy Questionnaire (KCCQ-CS)and death from all causes [99,100,101,102,107]. In female patients with HFpEF, Sac/Val reduces systolic blood pressure (SBP) which is directly associated with the N-terminal brain natriuretic peptide (NT-proBNP) level. However, the reduction in SBP would not account for the primary outcomes since regardless of sex, there was no difference [99]. Recently, Sac/Val has been shown to reduce plasma levels of NT-proBNP in patients with HFpEF of both sexes compared to standard treatment with an RAS inhibitor (ACE inhibitor, enalapril or ARB, valsartan) [100,101]. The reduced NT-proBNP was associated with a lower risk of subsequent hospitalizations [100].

Lately, US Food and Drug Administration has expanded the clinical use of empagliflozin, an sodium/glucose cotransporter 2inhibitor (SGLT2i), which reduced the risk of cardiovascular death and HF hospitalization [108] in a cohort including only about 24% women [103]. The most recent guidelines from the American College of Cardiology/American Heart Association (ACC/AHA) give SGLT2 inhibitors a class 2a recommendation in the treatment of HFpEF. Another SGLT2i, dapagliflozin [109], regardless of sex, reduces the risk of worsening HF events or death consequent to CVD [104].

SGLT2i, empagliflozin, dapagliflozin, canagliflozin, and ertugliflozin play the potential role of anti-HFpEF through direct or indirect synergy of multiple targets and pathways. The synergism could occur with targets and signaling pathways involved in inflammation, vasculature development, heart development, regulation of the MAPK cascade, ion transport, cell proliferation, apoptosis, oxidative stress, cell adhesion, upregulation of cell death, growth factor response and cell response to lipids [110].

In cardiomyocytes, isolated from 30 patients with HFpEF, empagliflozin, a SGLT-2i, significantly suppresses the increased levels of intercellular adhesiom molecular 1 (ICAM-1), vascular cell adhesion molecule 1, TNF-α and IL-6, andattenuates the parameters of pathological oxidative stress.HFpEF induces an increase in oxidized PKG1a, which appears as dimers in the outer membrane of cardiomyocytes, due to eNOS activation consequent to oxidative stress. Empaglifozin increases PKG1a monomers translocated back to the cytosol through the increase in the concentration of cGMP [111]. Thus, empaglifozin could be a therapeutic strategy to improve pathological cardiac stiffness in patients with HFpEF.

SGLT2i is associated with a lower incidence of primary outcomes such as first hospitalization for heart failure or death from cardiovascular events in patients with HFpEF. However these beneficial effects appear to be combined with an increased risk of urinary tract infections [112]. Dapaglifozin, another SGLT2i, improves KCCQ-CS primary as well as secondary endpoint and promoted weight reduction. Dapalgifozin improves symptoms such as exercise limitations and is well tolerated by patients with HFpEF [105].

Since there is an inflammatory basis for the development of HFpEF, the use of an IL-1 blocker could be an excellent therapeutic target. Blockade of IL-1 by anakinra (recombinant IL-1 receptor antagonist) reduced plasma levels of CRP and NT-proBNP in patients with HFpEF. However, it did not improve aerobic exercise capacity or ventilation efficiency [106].

Randomized trials using beta-blockers for the treatment of HFpEF are still few and in general, the available results demonstrate only a low risk of all-cause mortality regardless of sex [96]. Furthermore, the evaluation of the use of beta-blockers for the treatment of HFpEF has been quite limited in patients. The meta-analysis conducted in 2020 identified only 5 randomized clinical trials, including only 538 patients, regardless of gender. Results indicated that beta-blockers did not significantly alter NYHA class, exercise capacity expressed as metabolic equivalents or plasma levels of B-type natriuretic peptide (BNP). Thus, no clear beneficial effect of beta-blockers on the severity of HFpEF was found [113]. However, according to findings in preclinical trials, the sympathetic nervous system is closely involved in the development of menopause-induced HFpEF, thus showing the importance of a well-designed study for this specific patient group.

### 3.2. Hormone Replacement Therapy

Clinical trials of HRT and treatment of HFpEF still yield conflicting results. It is the complex hormonal environment that affects cellular and organ functions involved in the development and progression of CVD in ageing women. The attempt to combine conjugated estrogen derived from horse urine with a progesterone endocrine disruptor, medroxyprogesterone acetate, does not produce the comparable beneficial effects of human-identical female endogenous hormones [114]. This formulation showed important adverse effects such as increased blood clotting and inflammatory markers when compared to the use of transdermal estradiol [115]. It is likely that the failure to achieve cardiac benefits is related to the use of foreign agents to the human female body and the timing of beginning of therapy [19,116]. Recently, a significant reduction of CAD, breast cancer, hip fracture and all-cause mortality was observed, when conjugated equine estrogens were administrated in women. In contrast, the combination of estrogen and medroxyprogeterone acetate did not reduce the overall risk, but also increased the risk of breast cancer [117].

### 3.3. Phytoestrogen

There are absolute contraindications to HRT, including undiagnosed abnormal vaginal bleeding, active thromboembolic disorder or acute phase myocardial infarction, suspected or active breast or endometrial cancer, and active liver disease with liver tests showing abnormal liver function [118]. Thus, an alternative is phytoestrogens (PhytoE) [119,120]. However, the currently available data regarding the use of phytoestrogens for the treatment of CVD are still controversial.

PhytoE are estrogenic compounds of plant origin, and the group that has the highest amount of PhytoEare isoflavones and lignans [120,121,122]. Among the PhytoEmost studied for cardiovascular effects are daidzein and genistein, and the results are controversial regarding cardiac benefits [123]. Reduced serum levels of glycitin and genistein would be associated with an increased risk of cardiac events and absence of glycitin was associated with earlier hospitalization for angina [123].

A meta-analysis was performed to determine the effects of PhytoEsupplementation on intermediate CVD risk factors in postmenopausal women. Use of PhytoEwas associated with a decrease in serum total cholesterol, low-density lipoprotein, triglycerides, apolipoprotein B and in serum ICAM-1 and E-selectin and with an increase in serum apolipoprotein A-1. However, the use of PhytoE appears to have modest benefits on the risk of CVD, but may be a risk factor for atherosclerosis due to the possible increase in the thickness of the carotid intima [124]. Similarly, genistein supplementation interferes positively on lipid profile, blood pressure, body mass index and body weight. Genistein significantly reduced total cholesterol (TC), low-density lipoprotein cholesterol (LDL-C) and SBP with no change in the other parameters. In addition, ingestion of genistein for more than 6 months showed a greater effect in reducing cholesterol and DBP but induces an increase in triglycerides (TG) and LDL-C [125].

Since PhytoEare present at large scales in some grains, a meta-analysis study was recently conducted for the use of flaxseed (*Linum usitatissimum* L.), soybean (*Glycine max* L.) and red clover (*Trifoliump ratense* L.) aimed at the treatment of CVD risk factors [126]. Flaxseed contains phytochemicals, such as the lignan complex: secoisolariciresinoldiglycoside, cinnamic acid glycoside and hydroxymethylglutaric acid [127], which would be involved in the suppression of inflammatory tissue damage caused by oxidative stress, reduction of total cholesterol, modulation of 7α-hydroxylase and acyl enzymes -coenzyme A: cholesterol acyltransferases and reduction of arachidonic acid production and, consequently, reduction of the inflammatory process [126]. Red clover grain has a composition of isoflavones with higher concentrations of formononetin and biochanin A and lower concentrations of daidzein and genistein [128]. These show similarities with endogenous 17β-estradiol and show greater affinity for ER-β [129]. Flaxseed intake by postmenopausal women is associated with a significant reduction in TC, LDL-C, and high-density lipoprotein cholesterol levels density (HDL-C). The effect of soy protein on the lipid profile showed a significant decrease in TC and LDL-C levels, as well as a significant increase in HDL-C levels. Changes in lipid profile showed a significant reduction in TC levels after using red clover and a significant increase in HDL-C levels. Consumption of flaxseed, soy and red clover may have a beneficial effect on lipids in postmenopausal women and suggests a favorable effect in preventing CVD [129].

*Elaeagnus angustifolia* L., known as Senjed in Persian, contains two types of PhytoEincluding β-sitosterolandstigmometol [130]. The effects of this medicinal plant were evaluated in a trial including 58 postmenopausal women (45–70 years) and demonstrated a reduction in heart rate and serum levels of LDL-C and HDL-C after ten weeks of treatment with *E. angustifolia*, indicating possible beneficial effects in this group of patients. However, the placebo (isomalt and corn starch) used in this trial also showed significant effects [131].

### 3.4. Lifestyle Change and Healthy Diet

Improving quality of life is a recurring theme in several diseases, including CVD. The Women’s Health Initiative Randomized Controlled Dietary Modification (WHIRCDMT) study tested whether the Dietary Guidelines for Americans (US Department of Agriculture) protected against CAD and other chronic diseases. Postmenopausal women with CVD randomized to a ‘heart healthy’ low-fat diet in 1993 had a 26% increased risk of developing additional heart events [132]. In 2017 [133], the WHIRCDMT included data from an additional 5 years of follow-up indicating that the risk of CAD in this subgroup of postmenopausal women increased even further from 47% to 61%. The unfavorable outcomes observed in postmenopausal possibly occur because women were less likely to adhere to the prescribed dietary intervention. In addition, the increased evolution to CAD could be explained by a direct consequence of postmenopausal characteristics of insulin resistance [134], which is the most powerful predictor of future development of CVD in women. These characteristics lead to the importance to prescribe the heart-healthy low-fat, high-carbohydrate diet, at least in postmenopausal women.

It is essential to combine established cholesterol-lowering foods (vegetable protein, nuts, viscous fiber and phytosterols) and monounsaturated fat to promote a clinically significant reduction in lipid levels to prevent CVD [135]. Recently, in a prospective cohort study of postmenopausal women in the United States, greater adherence to the diet was associated with a lower risk of total CVD (11%), CAD (14%) and HF (17%) [136]. A lower risk of developing HFpEF and HFrEF among postmenopausal women is associated with having one or more healthy lifestyle habits, e.g., eating a healthy diet, being physically active, not smoking and avoiding being overweight [137,138].

An intense level of recreational physical activity, including walking, is associated with a significant reduced risk of HF in older women. Total physical activity but not high-intensity correlates inversely with the appearance of HF. This suggests that the volume of physical activity (e.g., movement) rather than intensity may be the factor underlying the exercise benefit [139].

## 4. Conclusions

Despite all the advances in considering the best options to treat menopause-induced HFpEF, which involves the reduction of cardioprotection due to estrogen depletion, the focus should be the modulation of the RAS and sympathetic system. Clinical trials for new therapeutic approaches are not yet enlightening to the point of providing new treatment guidelines for this particular group of patients. However, the combination of the alternatives described above and lifestyle changes including physical activity and healthy eating seem to be associated with a better prognosis for HFpEF.

## Figures and Tables

**Figure 1 ijms-23-15140-f001:**
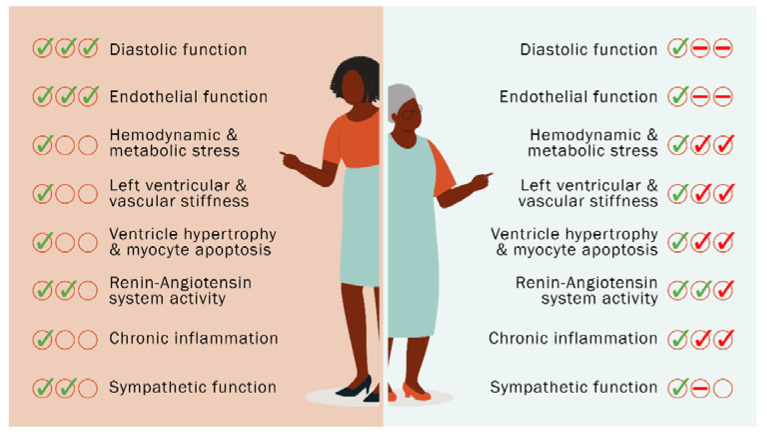
Cardiovascular parameters of premenopausal healthy women (left panel) are altered after menopause (right panel) and underlie the increased risk of HFpEF and mortality in this population, with either reduction (red dashes) or pathological exacerbation (red checks).

**Figure 2 ijms-23-15140-f002:**
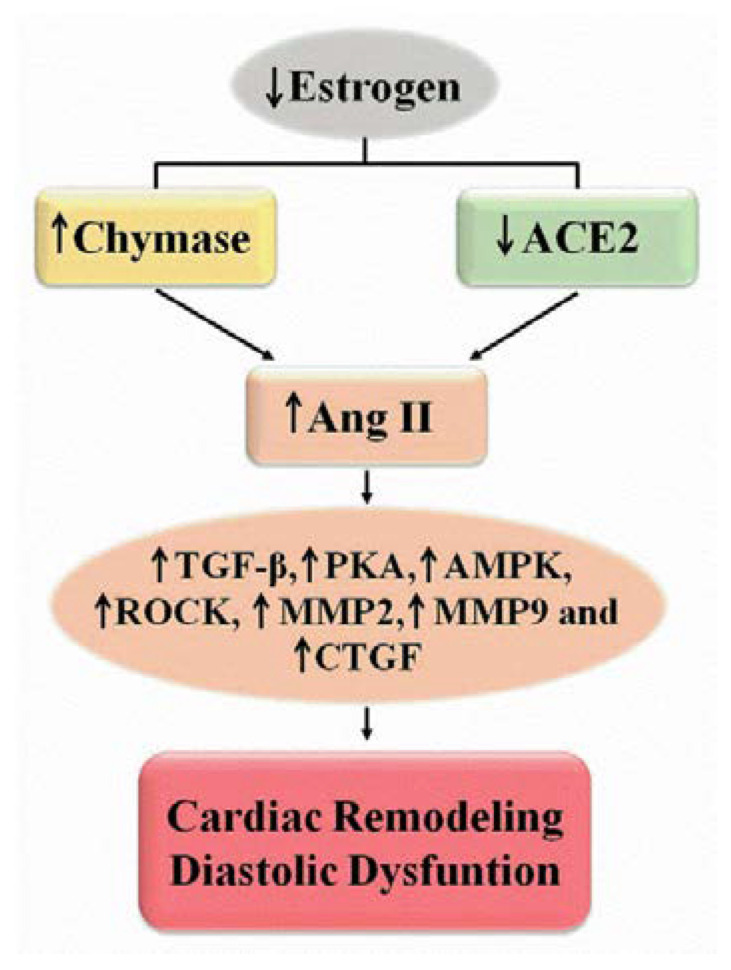
Estrogen depletion induces increased cardiac Ang II. ACE2—angiotensinogen-converting enzyme 2; AMPK—AMP kinase; CTGF—connective tissue growth factor; MMP—matrix metalloproteinase; PKA—protein kinase A; ROCK—Rho kinase; TGF-β—transforming growth factor beta.

**Figure 3 ijms-23-15140-f003:**
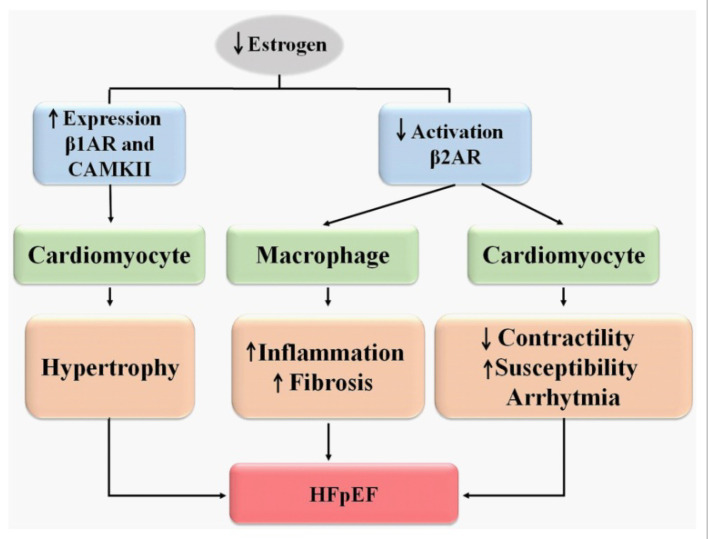
Estrogen depletion modulating beta-adrenergic receptors. β1AR—beta 1 adrenergic receptors; β2AR—beta 2 adrenergic receptors; HFpEF—heart failure with preserved ejection fraction.

**Table 1 ijms-23-15140-t001:** Alternative therapies for HFpEF.

Drugs	Response	% Women in Clinical Trial	References
Perindoprol, candesartan, irbesartan, spironolactone, sacubitril/valsartan (Sac/Val)	Mostly neutral response	-	[10,96]
Spironolactone	Neutral response between the sexes: ↓ All-cause mortality in women	49.9	[97]
Sac/Val	Potential benefit in women	≈50	[98]
Sac/Val	↓ SBP more in women than in men ↓ Plasma levels of NT-proBNP No response on primary outcome	52	[99]
Sac/Val	↓ Plasma levels of NT-proBNP in both sexes	50	[100,101]
	↓ Greater risk of hospitalization for HF in women than in men		[100]
5Sac/Val	↓ Primary outcome in the fragilest patient ↓↓ Secondary clinical outcomes for most fragile patients	50	[102]
Empagliflozin	↓ Risk of cardiovascular death and HF hospitalization	24	[103]
Dapagliflozin	↓ Risk of worsening HF events or death regardless of sex	44	[104]
Dapaglifozin	Improves primary endpoint and secondary endpoint ↓ Weight	57	[105]
Anakinra	↓ Plasma levels of CRP and NT-proBNP	64	[106]

CRP—reactive protein; CVD—cardiovascular disease; NT-proBNP—N-terminal brain natriuretic peptide; Sac/Val—sacubitril/valsartan; SBP—systolic blood pressure.

## Data Availability

Not applicable.

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
