# Peer review of "Heart Failure in Menopause: Treatment and New Approaches"

_ijms, 2022, doi:10.3390/ijms232315140_

Round 1

Reviewer 1 Report

This review explores the alterations observed in the condition of HFpEF induced by menopause and the therapeutic targets with potential to interfere with the disease progress. I have some concerns.

1. This review is about treatment and new approaches of heart failure in menopause. However, only a small amount of the contents are focusing on the treatment. The contents in the 3rd part (Treatment of women with menopausal-induced HF) should be increased, while the contents in the 1st part and the 2nd part should be decreased.

2. More specific treatments for menopausal-induced HF should be added into the 3rd part (Treatment of women with menopausal-induced HF).

3. The treatments in the 3rd part should be based on the 2nd part (Mechanism of menopause-induced HFpEF and targets for treatment). There are large amount of the contents in the 2nd part about the sympathetic nervous system (2.2.1). However, this didn’t contribute anything in the 3rd part.

4. The development of cardiovascular disease has been attributed to the reduced level of circulating estrogen during menopause. Why is the incidence of CVD in men lower than women since men never have estrogen? This should be clarified.

Author Response

This review explores the alterations observed in the condition of HFpEF induced by menopause and the therapeutic targets with potential to interfere with the disease progress. I have some concerns.

  1. The reviewer commented: “this review is about treatment and new approaches of heart failure in menopause. However, only a small amount of the contents are focusing on the treatment. The contents in the 3rd part (Treatment of women with menopausal-induced HF) should be increased, while the contents in the 1st part and the 2nd part should be decreased.

As suggested by the reviewer, the text regarding treatment was expanded.

The following paragraph was included to the subtitle of treatment:

Pages 10-11:

Modulation of the RAS for the treatment of HFpEF has shown mostly neutral results for drugs such as perindoprol, candesartan, irbesartan, spironolactone, and sacubitril/valsartan (Sac/Val)[10,104]. In regard of the analysis of the Aldosterone Antagonist Therapy for Adults with Heart Failure and Preserved Systolic Function (TOPCAT - NCT00094302) there were no differences between the sexes using placebo or spironolactone for the primary outcome or its components. However, there was a reduction in all-cause mortality associated with spironolactone therapy in women, with significant interaction between sex[105]. A comparative study between the use of valsartan and Sac/Valin patients with HFpEF in which approximately 50% were female showed a potential benefit for this specific group[106]. The major clinical studies related to the use of Sac/Val, for the treatment of HFpEF, evaluate: 1. primary outcome as a composite of total hospitalizations (first and recurrent) and death from cardiovascular causes, myocardial infarction, or stroke, all-cause mortality, and renal outcome (decreased glomerular filtration, development of end-stage renal disease, or death from renal failure); 2. secondary outcome including change in Clinical Summary Score of the Kansas City Cardiomyopathy Questionnaire (KCCQ-CS) and death from all causes[107–110,115]. In female patients with HFpEF, Sac/Val reduces systolic blood pressure (SBP) which is directly associated with the N-terminal brain natriuretic peptide (NT-proBNP) level. However, the reduction in SBP would not account for the primary outcomes since regardless sex, there was no difference [107]. Recently, Sac/Val has been shown to reduce plasma levels of NT-proBNP in patients with HFpEF of both sexes compared to standard treatment with an RAS inhibitor (ACE inhibitor, enalapril or ARB, valsartan)[108,109]. The reduced NT-proBNP was associated with a lower risk of subsequent hospitalizations[108].

Lately, US Food and Drug Administration has expanded the clinical use of empagliflozin, an sodium/glucose cotransporter 2 inhibitor (SGLT2i), which reduced the risk of cardiovascular death and HF hospitalization [116] in a cohort including only about 24% women[111]. The most recent guidelines from the American College of Cardiology/American Heart Association (ACC/AHA) give SGLT2 inhibitors a class 2a recommendation in the treatment of HFpEF. Another SGLT2i, dapagliflozin[117],regardless of sex, reduces the risk of worsening HF events or death consequent to CVD[112].

SGLT2i, empagliflozin, dapagliflozin, canagliflozin, and ertugliflozin play the potential role of anti-HFpEF through direct or indirect synergy of multiple targets and pathways. The synergism could occur with targets and signaling pathways involved in inflammation, vasculature development, heart development, regulation of the MAPK cascade, ion transport, cell proliferation, apoptosis, oxidative stress, cell adhesion, upregulation of cell death, growth factor response and cell response to lipids[118].

In cardiomyocytes, isolated from 30 patients with HFpEF, empagliflozin, a SGLT-2i, significantly suppresses the increased levels of intercellular adhesiom molecular 1 (ICAM-1), vascular cell adhesion molecule 1, TNF-α and IL-6, and attenuates the parameters of pathological oxidative stress. HFpEF induces increase of oxidized PKG1a, which appears as dimers in the outer membrane of cardiomyocytes, due to eNOS activation consequent to oxidative stress. Empaglifozin increases PKG1a monomers translocated back to the cytosol through the increase in the concentration of cGMP[119]. Thus, empaglifozin could be a therapeutic strategy to improve pathological cardiac stiffness in patients with HFpEF.

SGLT2i is associated with a lower incidence of primary outcomes such as first hospitalization for heart failure or death from cardiovascular events in patients with HFpEF. However, these beneficial effects appear to be combined with an increased risk of urinary tract infections[120]. Dapaglifozin, another SGLT2i, improves KCCQ-CS primary endpoint, as well as secondary endpoint and promoted weight reduction. Dapalgifozin improves symptoms such as exercise limitations and is well tolerated by patients with HFpEF[113].

Since there is an inflammatory basis for the development of HFpEF, the use of an IL-1 blocker could be an excellent therapeutic target. Blockade of IL-1 by anakinra (recombinant IL-1 receptor antagonist) reduced plasma levels of CRP and NT-proBNP in patients with HFpEF, however, it did not improve aerobic exercise capacity or ventilation efficiency[114].

Randomized trials using beta-blockers for the treatment of HFpEF are still few and in general the available results demonstrate only a low risk of all-cause mortality regardless of sex[104]. Furthermore, the evaluation of the use of beta-blockers for the treatment of HFpEF has been quite limited in patients, the meta-analysis carried out in 2020 identified only 5 randomized clinical trials, including only 538 patients, regardless of gender, and indicated that beta-blockers did not significantly alter NYHA class, exercise capacity expressed as metabolic equivalents or plasma levels of BNP. Thus, no clear beneficial effect of beta-blockers on the severity of HFpEF was found[121]. However, according to findings in preclinical trials, the sympathetic nervous system is closely involved in the development of menopause-induced HFpEF, thus showing the importance of a well-designed study for this specific patient group.

  1. More specific treatments for menopausal-induced HF should be added into the 3rd part (Treatment of women with menopausal-induced HF).

More specific treatments including several clinical trials using the combination of sacubitril/valsartan, as well as SGLT2 inhibitors were included in the 3rd part as suggested by the reviewer. The treatments were summarized in table 1.

  1. The treatments in the 3rd part should be based on the 2nd part (Mechanism of menopause-induced HFpEF and targets for treatment). There are large amount of the contents in the 2nd part about the sympathetic nervous system (2.2.1). However, this didn’t contribute anything in the 3rd part.

Clinical trials using beta-blockers for the treatment of HFpEF are so far quite limited, especially with regard to postmenopausal HF (Fukuta et al in 2020). In this meta-analysis study, it was described only 5 randomized clinical trials including only 538 patients regardless of sex and no clear beneficial effect of beta-blockers on the severity of HFpEF was found (Fukuta et al., 2021). Yoon and Eom (2019) confirmed the lack of clinical evidences of the beneficial effects of beta-blockers for HFpEF. Authors chose not to remove the topic that addresses the sympathetic nervous system from the review, as according to findings in preclinical trials, the sympathetic nervous system is closely involved in the development of menopause-induced HFpEF, thus showing the importance of well-designed clinical study for this specific patient group.

References

Fukuta, H., Goto, T., Wakami, K., Kamiya, T., & Ohte, N. (2021). Effect of beta-blockers on heart failure severity in patients with heart failure with preserved ejection fraction: a meta-analysis of randomized controlled trials. Heart Failure Reviews, 26(1), 165–171. https://doi.org/10.1007/s10741-020-10013-5

Yoon, S., & Eom, G. H. (2019). Heart failure with preserved ejection fraction: present status and future directions. Experimental and Molecular Medicine, 51(12). https://doi.org/10.1038/s12276-019-0323-2

  1. The development of cardiovascular disease has been attributed to the reduced level of circulating estrogen during menopause. Why is the incidence of CVD in men lower than women since men never have estrogen? This should be clarified.

The prevalence, pathophysiology, and mortality rates associated with CVD differ between men and women according to age(Kuznetsova, 2018). Premenopausal women have lower CVD prevalence and mortality compared to men, while postmenopausal CVD prevalence becomes similar in both sexes(Kuznetsova, 2018), directly related to the reduction of cardioprotective effect of estrogen. Because of higher life expectancy of women, the prevalence of CVD is equal in male and female elderly population but with higher mortality in women due to low level of estrogen. It is usual the initial appearance of coronary heart disease in men, while women have stroke or HF as the first event (Leening et al., 2014). These differences in CVD incidence between men and women decrease with increasing age(Louis et al., 2019), coinciding with estrogen depletion observed in post-menopausal women.

The text above has been included to the manuscript.

References

Crandall, J. P., Oram, V., Trandafirescu, G., Reid, M., Kishore, P., Hawkins, M., Cohen, H. W., & Barzilai, N. (2012). Pilot Study of Resveratrol in Older Adults With Impaired Glucose Tolerance. The Journals of Gerontology: Series A, 67(12), 1307–1312. https://doi.org/10.1093/gerona/glr235

Kuznetsova, T. (2018). Sex Differences in Epidemiology of Cardiac and Vascular Disease. In Advances in experimental medicine and biology (Vol. 1065, pp. 61–70). https://doi.org/10.1007/978-3-319-77932-4_4

Leening, M. J. G., Ferket, B. S., Steyerberg, E. W., Kavousi, M., Deckers, J. W., Nieboer, D., Heeringa, J., Portegies, M. L. P., Hofman, A., Ikram, M. A., Hunink, M. G. M., Franco, O. H., Stricker, B. H., Witteman, J. C. M., & Roos-Hesselink, J. W. (2014). Sex differences in lifetime risk and first manifestation of cardiovascular disease: prospective population based cohort study. BMJ, 349(nov17 9), g5992–g5992. https://doi.org/10.1136/bmj.g5992

Louis, X. L., Raj, P., Chan, L., Zieroth, S., Netticadan, T., & Wigle, J. T. (2019). Are the cardioprotective effects of the phytoestrogen resveratrol sex-dependent? Canadian Journal of Physiology and Pharmacology, 97(6), 503–514. https://doi.org/10.1139/cjpp-2018-0544

Reviewer 2 Report

Lines of the manuscript are not numbered, so it is difficult to exactly indicate the place for changes.

3rd and 4th paragraphs, page 2, needs to be referenced. Please check each paragraph to be properly referenced in the entire manuscript.

At the final of Introduction section, the aim of the study must be presented, detailing also the novelty/special aspects that your research brings to the field and the reason for you have chosen this topic (as there are plenty of studies in the field, developing similar reviews.

As the Instructions for authors require, Abbreviations should be defined the first time they appear in each of 3 sections: the abstract; the main text; under the first figure or table. When defined for the first time, the acronym/abbreviation/initialism should be added in parentheses after the written-out form. Revise the entire manuscript in this regard.

Adjust the figures’ size to the text, they are huge in the actual shape.

Two subchapters need to be added to your article, in order to complete the frame of your topic. 

Please make a novel subchapter where you discuss the potential beneficial effects of estrogenic supplementation on the cardiovascular function of women, in the stages of menopause. I suggest cheeking and referring to Tit D.M., Pallag A., Iovan C., Furau G., Furau C., Bungau, S. Somatic-vegetative symptoms evolution in postmenopausal women treated with phytoestrogens and hormone replacement therapy. Iran. J. Public Health 2017, 46(11), 1128-1134

Also, please discuss in a separate subchapter the particularities of heart failure treatment in women, in menopause; if there were any particular findings in novel studies that evaluated sacubitril/valsartan or Sglt2-inhibitors, statins, nano treatments, etc. I suggest checking and referring to https://doi.org/10.3390/diagnostics10070483; https://www.mdpi.com/1422-0067/23/19/11336; https://doi.org/10.3390/chemosensors9040067 A summarising table would be relevant and would add a table to your paper (tabulating part is totally missing). The last column of the suggested table must be Ref. (references)

Author Response

  1. Lines of the manuscript are not numbered, so it is difficult to exactly indicate the place for changes. 3rdand 4th paragraphs, page 2, needs to be referenced. Please check each paragraph to be properly referenced in the entire manuscript.

As commented by the reviewer, references have been included to the indicated paragraphs.

  1. At the final of Introduction section, the aim of the study must be presented, detailing also the novelty/special aspects that your research brings to the field and the reason for you have chosen this topic (as there are plenty of studies in the field, developing similar reviews.

As suggested by the reviewer, the following text has been included at the end of the introduction:

HFpEF in women has been related to the loss of cardioprotection due to the reduction in estrogen levels induced by menopause. Thus, this review seeks to indicate some of the mechanisms that are involved in HFpEF in women indicating possible targets for therapy, as well as addressing the most recent clinical approaches for this population

  1. As the Instructions for authors require, Abbreviations should be defined the first time they appear in each of 3 sections: the abstract; the main text; under the first figure or table. When defined for the first time, the acronym/abbreviation/initialism should be added in parentheses after the written-out form. Revise the entire manuscript in this regard.

All text has been revised.

  1. Adjust the figures’ size to the text, they are huge in the actual shape.

 Figures have been reformatted as suggested

  1. Two subchapters need to be added to your article, in order to complete the frame of your topic. Please make a novel subchapter where you discuss the potential beneficial effects of estrogenic supplementation on the cardiovascular function of women, in the stages of menopause. I suggest cheeking and referring to TitM., Pallag A., Iovan C., Furau G., Furau C., Bungau, S. Somatic-vegetative symptoms evolution in postmenopausal women treated with phytoestrogens and hormone replacement therapy. Iran. J. Public Health 2017, 46(11), 1128-1134. Also, please discuss in a separate subchapter the particularities of heart failure treatment in women, in menopause; if there were any particular findings in novel studies that evaluated sacubitril/valsartan or Sglt2-inhibitors, statins, nano treatments, etc. I suggest checking and referring to https://doi.org/10.3390/diagnostics10070483; https://www.mdpi.com/1422-0067/23/19/11336; https://doi.org/10.3390/chemosensors9040067

Authors considered the comments of the reviewer and the subchapters were added to expand the approach of use of sacubitril/valsartan and SGLT2 inhibitors in topic 3.1. In addition, a subtopic for phytoestrogen was also added in 3.2.

  1. A summarising table would be relevant and would add a table to your paper (tabulating part is totally missing). The last column of the suggested table must be Ref. (references)

As suggested by the reviewer, a table containing the summary of data e references has been included to the manuscript.

Round 2

Reviewer 1 Report

The authors have addressed my concerns.

Reviewer 2 Report

The authors made some improvements. However, the paper can be better developed, edited, and referenced. Please see my previous report and proceed consequently.